# Cognitive and Neural Differences in Exact and Approximate Arithmetic Using the Production Paradigm: An fNIRS Study

**DOI:** 10.3390/bs15010033

**Published:** 2025-01-01

**Authors:** Tianqi Yue, Buxuan Guan, Yan Wu

**Affiliations:** 1School of Psychology, Northeast Normal University, Changchun 130024, China; yuetq@nenu.edu.cn (T.Y.); guanbx708@nenu.edu.cn (B.G.); 2Dalian No.13 Senior High School, Dalian 116021, China

**Keywords:** production paradigm, exact arithmetic, approximate arithmetic, fNIRS

## Abstract

This study investigated the cognitive and neural mechanisms of exact and approximate arithmetic using fNIRS technology during natural calculation processes (i.e., the production paradigm). Behavioral results showed (1) a significantly longer reaction time for exact arithmetic compared to approximate arithmetic, and (2) both exact and approximate arithmetic exhibited a problem size effect, with larger operands requiring more time. The fNIRS results further revealed differences in the neural bases underlying these two arithmetic processes, with exact arithmetic showing greater activation in the L-SFG (left superior frontal gyrus, CH16), while approximate arithmetic exhibited problem size effect in the right hemisphere. Additionally, larger operands registered more brain activities in the R-DLPFC (right dorsolateral prefrontal cortex, CH4), R-SFG (right superior frontal gyrus, CH2), and PMC and SMA (pre- and supplementary motor cortexes, CH3) compared to smaller operands in approximate arithmetic. Moreover, correlation analysis found a significant correlation between approximate arithmetic and semantic processing in the R-PMC and R-SMA (right pre- and supplementary motor cortexes). These findings suggest a neural dissociation between exact and approximate arithmetic, with exact arithmetic processing showing a dominant role in the left hemisphere, while approximate arithmetic processing was more sensitive in the right hemisphere.

## 1. Introduction

Calculation ability is an essential mathematical skill frequently used to solve everyday problems such as calculating the total price while shopping or summing up the sales of various products. This process of performing exact calculations to obtain accurate results is known as exact arithmetic ([35]). However, when determining whether a purchase exceeds 100 yuan or estimating the expenses of a project, an approximate result is often sufficient. This process is referred to as approximate arithmetic ([36]). Exact and approximate arithmetic are commonly used daily, but do they involve the same cognitive processing? Although previous studies have explored this question, differing perspectives remain ([11]; [14]; [28]; [42]; [45]). [28] ([28]) found that both children and adults exhibit no significant neural dissociation between exact and approximate arithmetic, a result that does not fully align with the neural separation model proposed by [14] ([14]). This suggests that these tasks may involve overlapping cognitive processes influenced by individual strategies and task demands. Other studies have approached this issue from a developmental perspective. For example, [11] ([11]) found that children activate neural networks similar to those of adults during exact and approximate arithmetic but at lower activation levels, relying more heavily on visual space strategies. There have also been studies that have only explored the activation patterns of the brain in relation to exact arithmetic ([34]; [50]).

In contrast, research specifically investigating the neural mechanisms underlying multi-digit exact and approximate arithmetic remains relatively scarce. Additionally, previous research paradigms often lacked ecological validity, failing to explore calculation processes in naturalistic settings. Thus, the aim of the current study is to investigate the cognitive processing and neural mechanisms of exact and approximate arithmetic using the production paradigm. Addressing this issue can enhance our understanding of the advanced cognitive process of mathematical calculation and provide valuable insights for teaching approximate and exact arithmetic in primary and secondary education.

Arithmetic encompasses operations such as addition, subtraction, multiplication, division, and various complex mixed operations. Although both exact and approximate arithmetic are types of mathematical processing, previous research has identified differences in their underlying mechanisms. [18] ([18]) investigated the developmental trends of exact and approximate two-digit multiplication in fourth-, fifth-, and sixth-grade children and university students. Results indicated that accuracy in exact arithmetic tasks increased with age during childhood, while accuracy in approximate arithmetic tasks peaked in the fourth grade and did not change significantly with age. This suggests that exact and approximate arithmetic reflect, to a certain extent, two different skills.

Furthermore, studies have shown that children who are more proficient in estimation tend to perform better on standardized math tests and other assessments of mathematical ability and comprehension ([1]; [4], [5]; [39], [38], [40]; [41]; [51]). These studies not only demonstrate that approximate arithmetic skills are crucial for children’s development but also highlight that exact and approximate arithmetic play different roles in human cognitive development. This raises the question: Do differences in cognitive development ultimately lead to differences in brain function in adults? Addressing this question could advance our understanding of the mechanisms underlying exact and approximate arithmetic.

### 1.1. Primary Experimental Paradigms for Arithmetic

In studies currently exploring this question, researchers often use a delayed verification paradigm and a comparison task to minimize the interference of hand movements on neural signal collection. In the delayed verification paradigm, participants are first presented with an arithmetic problem (e.g., 4 + 5) and then with two candidate answers (e.g., 9 □ 7). They are asked to select the correct answer (exact arithmetic task) or the most plausible answer (approximate arithmetic task) to examine the processes of exact and approximate arithmetic ([14]; [42]; [44]). Notably, because the delayed verification paradigm requires delayed responses, it is difficult to reveal the brain mechanisms underlying the immediate calculation process ([44]). The comparison task requires participants to estimate the result of a calculation problem displayed at the top of the screen and then determine whether that approximate result is larger or smaller than a given reference number, thereby examining the approximate arithmetic process ([2]; [17], [18]; [19]; [20]; [43]). However, this paradigm is more suitable for the approximate arithmetic task and lacks a corresponding exact arithmetic task format in terms of problem presentation. The third paradigm, known as the production paradigm, involves recording the cognitive and neural mechanisms during participants’ natural calculation processes. In the natural production paradigm, participants are directly presented with calculation problems without any options and are required to solve them as quickly and accurately as possible. They provide their answers either verbally or through input. Compared to the first two paradigms, the production paradigm is more natural and exhibits higher ecological validity, making it suitable for both exact arithmetic and approximate arithmetic tasks. Additionally, this approach avoids the potential influence of prompted answers inherent in the verification paradigm. However, due to the sensitivity of fMRI and ERP to hand movements, most existing studies investigating the neural mechanisms of exact and approximate arithmetic have relied on the other two paradigms.

### 1.2. Neural Mechanisms of Single-Digit and Multi-Digit Arithmetic

Using the delayed verification paradigm, [42] ([42]) first categorized single-digit numbers into large operands (5~9) and small operands (1~5) and used single-digit addition tasks (e.g., 5 + 6 for large operands, 1 + 2 for small operands) with fMRI and ERP techniques. Results revealed that exact arithmetic elicited greater activation in the left inferior frontal cortex (L-IFC) and bilateral angular gyrus (AG). In comparison, approximate arithmetic showed greater activation in the bilateral intraparietal sulcus (IPS), dorsolateral prefrontal cortex (DLPFC) and superior frontal cortex (SFC). These findings suggest distinct brain functions for exact and approximate arithmetic. Specifically, exact arithmetic is associated more with the left inferior frontal gyrus (L-IFG), cingulate gyrus (CG), and AG, which are closely linked to language processing and cognitive tasks and play a crucial role in automated fact retrieval. In contrast, approximate arithmetic is primarily reflected in the IPS, a region often regarded as central to quantity estimation and numerical processing. Based on the functional differences in the brain, the researchers inferred that exact arithmetic processing involves the retrieval of arithmetic facts (each exact arithmetic result stored in memory), while the approximation process primarily involves quantity processing ([42]).

However, unlike the aforementioned studies, two investigations suggest that the involvement of different brain regions may not only relate to the distinction between exact and approximate arithmetic but also to differences in operation types ([34]; [50]). [34] ([34]), using a verbal processing localization task, found that subtraction elicited greater activity in the IPS, whereas multiplication showed higher activation in regions associated with verbal processing, including the middle temporal gyrus (MTG) and IFG. This finding highlights potential neural network differences between arithmetic operations. Similarly, [50] ([50]) examined brain activation patterns elicited by single-digit addition and multiplication problems. Their results indicated neural network differences between the two operations, with addition relying more on visual space processing and multiplication depending more on verbal processing.

The above results seem to reveal that single-digit multiplication, but not necessarily addition, depends on language processing. The involvement of language processing in multiplication may be due to the rote memorization of multiplication tables in elementary school (especially among Chinese participants), resulting in the activation of language-related brain regions for single-digit multiplication ([50]). In a neuroimaging study examining changes in neural mechanisms after training ([15]), participants were trained to solve exact arithmetic problems, with post-training tests including both trained and untrained problems. The fMRI results indicated that post-training neural changes shifted from the IPS, which is sensitive to quantity processing, to the left angular gyrus (L-AG), which is associated with automated fact retrieval ([15]). However, the differences between single-digit exact and approximate arithmetic may be the result of training, while multi-digit calculations are difficult to achieve through training. Therefore, investigating multi-digit calculations is necessary.

In fact, prior research has shown that the brain mechanisms underlying multi-digit calculations are more complex. For example, one study (2018) compared three types of mathematical problems (i.e., number sequence completion, problem-solving, and geometric problem-solving) with exact multi-digit arithmetic operations (i.e., addition, subtraction, multiplication, and division) to exact arithmetic. Using fMRI, they found that solving complex mathematical problems generally activated all eight regions of the semantic system (AG, middle temporal gyrus (MTG), fusiform gyrus (FG), parahippocampal gyrus (PHG), dorsal medial prefrontal cortex (DMPFC), inferior frontal gyrus (IFG), ventral medial prefrontal cortex (VMPFC), and posterior cingulate gyrus (PCG)) more than exact arithmetic did. However, the exact arithmetic showed greater activation of the SMA and L-PCG, suggesting a stronger association with the brain’s language processing regions compared to complex mathematical problems, which indicates that different language processes are involved in mathematical tasks. However, [14] ([14]) found that in bilinguals, two-digit addition tasks in fMRI studies activated language-related brain regions such as the L-IFG, left cingulate gyrus (L-CG), and AG during exact arithmetic, whereas the approximate arithmetic process activated the IPS, postcentral sulcus (PCS), and inferior parietal lobule (IPL), which are associated with the visuospatial network. Additionally, [30] ([30]), using the verification paradigm and fMRI to study the activation patterns of multi-digit exact and approximate arithmetic, found that, compared to exact arithmetic, approximate arithmetic showed greater activation of the IFG, MTG, AG, and DMPFC, whereas exact arithmetic showed greater activation of the bilateral hippocampus (HPS) and left central sulcus (LCS).

Li et al. interpreted their findings from the perspectives of semantic and phonological processing, suggesting that the HPS and LCS are associated with phonological processing, which indicates greater phonological involvement in approximate arithmetic, whereas exact arithmetic relies on semantic processing, as indicated by the activation of related brain regions. Nevertheless, it is important to note that interpreting the differences between exact and approximate arithmetic solely on the functions of brain regions is somewhat subjective. The functions of each brain region are complex; for instance, some researchers believe that the IFG and MTG are involved in phonological processing, not just semantic processing ([34]). Therefore, relying only on the activated brain regions makes it impossible to determine which specific language network is involved in the processing of exact or approximate arithmetic.

### 1.3. Problem Size Effect in Single-Digit and Multi-Digit Arithmetic

In addition to the differences in brain mechanisms, single-digit and multi-digit calculations exhibit distinct problem size effects. The problem size effect refers to increased reaction times and error rates with larger operands ([23]). For example, solving 9 + 7 takes longer and is more prone to errors than solving 2 + 3. [42] ([42]) examined this effect in single-digit addition and subtraction tasks, measuring reaction times and accuracy using the delayed verification paradigm. They found that both exact and approximate single-digit calculations showed the problem size effect, with the effect more pronounced in exact arithmetic. The problem size effect in single-digit tasks was supported by several brain regions, including the bilateral IPS ([14]; [42]; [50]). However, the left intraparietal sulcus (L-IPS), left precentral sulcus (LPreCS), and L-IFG (Broca’s area) were more sensitive to the problem size effect in exact arithmetic ([42]). By contrast, [33] ([33]) used two-digit multiplication and the production paradigm to examine how reaction times change with the sum of the single-digits of the two operands in multi-digit exact and approximate arithmetic. He found that reaction times for exact arithmetic increased with the sum of the units’ digits of the operands, while reaction times for approximate arithmetic did not change. This suggests different processing modes for exact and approximate arithmetic in multi-digit tasks. However, most existing studies on problem size effects have focused on single-digit addition, subtraction, and multiplication, with limited research on multi-digit problem size effects, even though multi-digit arithmetic is more relevant to our daily lives.

### 1.4. The Current Study

In summary, although previous studies have explored exact and approximate arithmetic to some extent, significant research gaps remain in understanding the cognitive and neural mechanisms underlying these two types of calculations. Current studies on the neural mechanisms are still relatively limited, with ongoing debates about the specific brain regions involved and their relationship with language processing mechanisms. Most existing research focused on single-digit arithmetic using verification and comparison paradigms, which often emphasize controlled conditions rather than capturing the brain mechanisms involved in natural calculation processes. These paradigms lack ecological validity and may not reflect real-world multi-digit arithmetic. By contrast, the production paradigm used in this study can more accurately reflect individual calculation mechanisms in naturalistic settings. This approach, combined with fNIRS technology, offers the capability to investigate the cognitive and neural bases of exact and approximate arithmetic during natural calculation tasks.

Moreover, while previous studies have highlighted an association between arithmetic and language networks, particularly in single-digit tasks, the relationship between multi-digit arithmetic and language processing remains ambiguous. This study addresses these gaps by focusing on multi-digit arithmetic and its potential links to phonological and semantic processing. More importantly, this study recruited adults, not children, as participants. Unlike children, whose arithmetic processing reflects developmental trajectories, adults represent mature cognitive systems shaped by prolonged learning and practice. By focusing on adults, this study complements existing research that primarily examines children, bridging the gap between developmental and mature arithmetic processing. It offers new perspectives for a comprehensive understanding of arithmetic cognition and contributes to understanding how developmental differences between exact and approximate arithmetic ultimately shape distinct brain functions in adults. This provides both theoretical and practical implications for mathematics education and cognitive neuroscience.

Therefore, with the production paradigm, this study employed a 2 (exact arithmetic vs. approximate arithmetic) × 2 (large operands vs. small operands) within-subject experimental design using integer addition and subtraction to explore the cognitive-neural mechanisms of multi-digit exact and approximate arithmetic and their differences in an fNIRS experiment. Additionally, we conducted two language processing tasks, i.e., phonological and semantic tasks, to determine the association in brain activation between exact/approximate arithmetic and language processing. By adopting this design, we examined three questions: (1) What are the cognitive and neural bases for exact and approximate multi-digit calculations?; (2) What are the cognitive and neural differences between these two types of calculations?; and (3) What is the neural relationship between exact/approximate arithmetic and phonological or semantic processing, as indicated by overlapping patterns of brain activation? Given the limited research on multi-digit addition and subtraction, our predictions are based on studies of single-digit addition/subtraction or multi-digit multiplication ([14]; [42]). In this study, “neural relationships” refer to overlapping activation patterns between arithmetic and language-related tasks. This does not imply a direct functional overlap. First, we anticipated that the left superior frontal gyrus (L-SFG) would be associated with exact arithmetic, while regions in the right hemisphere, such as the R-DLPFC, R-SFG, R-PMC, and R-SMA, would be more sensitive to approximate arithmetic. Given that prior studies revealed a link between mathematical calculations and language processing ([14]; [27]; [50]), both exact and approximate arithmetic were hypothesized to be related to phonological and/or semantic processing. However, based on the results of [30] ([30]), we expected exact arithmetic to link more closely to phonological processing and approximate arithmetic to semantic processing.

## 2. Materials and Methods

### 2.1. Participants

We recruited 30 native Chinese-speaking college students with an average age of 21.6 years (Range = 18–24 years, SD = 1.62). After excluding participants with excessive artifacts detected during preprocessing, defined as losing more than 25% of signal channels, 29 valid data sets remained, consisting of 6 male and 23 female participants. All participants were right-handed, had normal or corrected-to-normal vision, no color blindness or color weakness, normal hearing, no history of psychiatric disorders, had not participated in similar experiments before, voluntarily signed informed consent forms, and received equal compensation after completing the experiment.

### 2.2. Experimental Design

The experiment utilized a 2 (arithmetic type: exact vs. approximate) × 2 (operand size: small vs. large) within-subject design. The dependent variable was the level of oxygenated hemoglobin activation under different conditions. Small and large operands refer to single-digit ± two-digit and two-digit ± two-digit operations, respectively.

### 2.3. Experimental Materials

The experiment consisted of four tasks, including exact and approximate arithmetic tasks and two language (i.e., phonological and semantic) processing tasks.

The problem sets for the exact and approximate arithmetic tasks each consisted of 30 single-digit ± two-digit problems and 30 two-digit ± two-digit problems, with half of the problems being addition and the other half subtraction. Considering known experimental effects in the mental arithmetic domain ([13]), the arithmetic problems were constructed under the following constraints (examples shown in Table 1): (1) No operands had zero as the units digit (e.g., 30 + 48); (2) No digits in the tens and/or units places were the same (e.g., 43 + 47 or 42 + 52); (3) No digits were repeated within a problem (e.g., 44 + 59); (4) There were no reverse order problems (e.g., if 52 + 76 is used, 76 + 52 is not); (5) There were no tie problems (e.g., 32 + 32); (6) Single-digit ± two-digit and two-digit ± two-digit problems resulted in a two-digit answer; (7) There were no sums with a single-digit of zero (e.g., 36 + 54); and (8) Subtraction problems were the reverse operations of addition problems (e.g., 26 + 52 → 78 − 52). 

Additionally, to control for differences between exact and approximate arithmetic tasks, the problems were counterbalanced. The same arithmetic problem was presented as an exact arithmetic task for half the participants and an approximate arithmetic task for the other half. The current study incorporates both addition and subtraction problems primarily because they are the most commonly used basic operations in daily life. Including both types of operations enhances the external validity of the study. Moreover, addition and subtraction exhibit structural symmetry in terms of operands, which helps balance different computational scenarios.

In the phonological processing task, participants were required to judge whether two presented single characters rhymed ([31]; [34]). To eliminate interference from visual information and ensure participants processed the visual and phonological information separately, the pairs of characters were categorized into four types: similar orthography and sharing the same rhyme (e.g., 粒/li4/and 泣/qi4/), similar orthography but different rhyme (e.g., 址/zhi3/and 扯/che3/), different orthography but sharing the same rhyme (e.g., 经/jing1/and 冰/bing1/), and different orthography and different rhyme (e.g., 绒/rong2/and 临/lin2/). To determine semantic relatedness, 34 university students rated the semantic association of the character pairs using a seven-point scale. Pairs with an average score below 3 (M = 2.43) were considered semantically unrelated. Additionally, a character-matching condition had participants judge whether two presented pseudo-characters (symbols created by rearranging the components of real characters) matched. Each condition included 12 trials, for a total of 60 trials.

The semantic processing task utilized the classic pyramid-shaped lexical semantic judgment task ([3]; [16]; [46]). Three two-character words were presented in a pyramid configuration, with a target word at the top of the screen and two options at the bottom. Participants judged which option (e.g., sun or shackle) was semantically related to the target word (e.g., handcuff). Additionally, a matching task involving three pseudo-words had participants identify which of the two options at the bottom of the screen was the mirror image of the target pseudo-word (as shown in Figure 1). Each participant completed 30 trials for each condition, for a total of 60 trials. The matching task served as a control condition to isolate visual interference in phonological and semantic processing. Using pseudographs devoid of phonological and semantic content established a baseline for visual matching demands, ensuring that responses in the phonological and semantic tasks were not confounded by visual similarity.

### 2.4. Experimental Procedure

The experimental procedure was programmed using E-Prime 2.0.10 and conducted on a computer with a screen resolution of 1920 × 1080 and a refresh rate of 60 Hz. The fixation point was white with a font size of 100. All experimental stimuli were image files, with both text and texture images measuring 80 px × 80 px. The experimental background was dark gray, and the stimuli were presented in black.

For the arithmetic tasks, exact and approximate arithmetic were presented in two separate blocks. Within each block, the conditions of operand size and addition/subtraction were randomized to prevent participants from guessing the purpose of the experiment. Participants performed exact and approximate arithmetic for the problems presented, input their results using the numeric keypad, and pressed the ENTER key to proceed to the next problem. Responses are recorded when the subject enters the first number. A random interval of 2–5 s was added between each trial. This study adopted a keyboard-based answer input design to simulate natural problem-solving scenarios closely. In real-life situations and multi-digit arithmetic tasks, individuals typically solve problems by generating answers rather than selecting from a limited set of options. The answer-generation paradigm using keyboard input avoids the potential cue effects introduced by multiple-choice formats, providing a more direct reflection of the problem-solving thought process. Additionally, since participants were current university students, keyboard input was a familiar and commonly used method for them. This approach not only aligns well with modern computerized educational settings but also achieves a balance between ecological validity and data collection accuracy. The workflow for the exact and approximate arithmetic tasks is shown in Figure 2. The order of exact and approximate tasks was counterbalanced among participants, with half of them performing the approximation task first, followed by the exact task, while the other half did the opposite. Practice trials were provided before the formal experiment. Each exact and approximate arithmetic task in the formal experiment included 60 trials, with 30 trials each for large and small operands and an equal number of addition and subtraction problems. Participants were allowed to take breaks between the two tasks to avoid fatigue effects. There are 15 practice opportunities before each task. Only after the participants reach the set accuracy standard in practice will they enter the formal experiment, ensuring that they fully understand the task rules.

In the phonological processing task, participants judged whether two sequentially presented Chinese characters rhymed or whether they matched. Each character was displayed for 800 ms, separated by a 200 ms blank interval. Afterward, a red fixation point (+) appeared, indicating that participants should respond. Each trial had a random interval of 2200 ms, 2600 ms, or 3000 ms.

In the semantic processing task, a fixation point was presented for 500 ms, followed by the simultaneous presentation of three experimental stimuli (three two-character words or three pseudo-words) for a maximum of 2500 ms. The stimuli disappeared after the participant pressed the response key. Finally, a blank screen appeared for approximately 500 ms. The random interval between trials ranged from 2 to 5 s. The workflow for the phonological and semantic processing tasks is shown in Figure 3.

### 2.5. fNIRS Data Collection

Data were collected using a functional near-infrared spectroscopy (fNIRS) brain imaging system (LABNIRS system) from Shimadzu, Japan, with a sampling rate of 19 times per second. The system included 16 sources and 16 detectors, with a 3 cm spacing between each detector and receiver in each channel. Previous research has shown that brain regions associated with language processing are predominantly located in the left hemisphere ([3]; [13]; [16]; [31]; [34]). Therefore, we positioned most of the sensors over the left side of the brain; the bilateral fronto-parietal network, left frontal lobe, and L-AG were selected as regions of interest, as shown in Figure 4. After completing the experiment, a 3D digitizer (FASTRAK, Polhemus, Colchester, VT, USA) was used to locate the coordinates of each channel, calculate the corresponding three-dimensional information (MNI coordinates), and identify the brain regions corresponding to each channel.

### 2.6. Data Analysis Methods

#### 2.6.1. Behavioral Analysis Methods

Reaction times were analyzed using a 2 × 2 repeated measures ANOVA in SPSS 24 (IBM SPSS Statistics), with Bonferroni post hoc tests. Collected data were preprocessed to remove erroneous trials, and only correct trials were included in the data analysis. For the approximate arithmetic task, the criteria proposed by [32] ([32]) were used, where estimates within 40% of the exact arithmetic result were considered reasonable. Specifically, if the percentage of the estimation result divided by the exact result was less than 40%, the trial was included in the analysis. This standard was applied in the current study, with at least 80% of the estimation trials retained. Following this standard trial exclusion process, an average of 2.89 trials (SD = 2.27) were excluded in the exact arithmetic condition, and an average of 8.58 trials (SD = 6.26) were excluded in the approximate arithmetic condition. The partial eta squared (η_p_^2^) effect size was reported, with values between 0.01 and 0.06 indicating a small effect, values between 0.07 and 0.14 indicating a medium effect, and values of 0.15 or higher indicating a large effect ([8]).

#### 2.6.2. fNIRS Analysis Methods

Data were preprocessed and analyzed using the NIRS_SPM package (based on Matlab) ([25]). The specific analysis methods were as follows. First, the raw oxygenation data were preprocessed using the hemodynamic response function (HRF) filtering, wavelet-minimum description length (wavelet-MDL) analysis, and Principal Component Analysis (PCA) to remove artifacts and global physiological noise, and correct signal distortions caused by factors such as breathing, heartbeat, vascular movement, and machine noise. Next, using each experimental condition as a regressor, the HbO signal changes were treated as predictors, and the beta values (i.e., dependent variables) for each condition were calculated. Finally, the beta values for each channel were analyzed using a 2 × 2 repeated measures ANOVA in SPSS 24 (IBM SPSS Statistics).

For the language task oxygenation data, channels showing significant differences between the rhyming task condition and the matching task condition in the phonological task were identified as regions of interest (ROIs) for correlation analysis. Similarly, channels showing significant differences between the semantic relatedness task condition and the pseudo-word-matching task condition in the semantic task were identified as ROIs for correlation analysis.

For the arithmetic task oxygenation data, channels showing significant differences between the exact and approximate arithmetic conditions were identified as ROIs for correlation analysis. The oxygenation data from these arithmetic task ROIs were then correlated with the oxygenation data from the language task ROIs identified during the arithmetic tasks. Specifically, phonological task ROIs were correlated with exact arithmetic task ROIs, and semantic task ROIs were correlated with approximate arithmetic task ROIs. Additionally, the beta values from the exact and approximate arithmetic conditions in the identified ROIs were correlated with the beta effect sizes from the phonological and semantic tasks.

The beta effect size for the phonological task was defined as the beta value of the rhyming task condition minus the beta value of the character matching task condition for each channel. Similarly, the beta effect for the semantic task was defined as the beta value of the semantic relatedness task condition minus the beta value of the pseudo-word-matching task condition for each channel. The purpose of subtracting the β values was to eliminate the influence of graphene processing, ensuring that the β effect sizes exclusively reflected language processing.

The beta effect size for exact/approximate arithmetic was defined as the beta value of the larger operand condition minus the beta value of the smaller operand condition for each channel. Subtracting the β values was used to assess the extent of involvement in the exact/approximate arithmetic task. Then, the beta effect size for exact/approximate arithmetic was correlated with the beta effect from the phonological and semantic tasks using Pearson correlation analysis.

## 3. Results

### 3.1. Behavioral Results

A 2 (arithmetic type: exact vs. approximate) × 2 (operand size: small vs. large) repeated measures ANOVA was conducted on the reaction times for the exact and approximate arithmetic tasks. Results showed a significant main effect of arithmetic type (*F* (1, 28) = 42.05, *p* < 0.001, η_p_^2^ = 0.6); a significant main effect of operand size (*F* (1, 28) = 192.12, *p* < 0.001, η_p_^2^ = 0.873); and a significant interaction between arithmetic type and operand size (*F* (1, 28) = 72.90, *p* < 0.001, η_p_^2^ = 0.723). Simple effects analysis revealed that the problem size effect was significant for both exact and approximate arithmetic, with reaction times significantly longer for large operands than for small operands. However, this difference was more pronounced for exact arithmetic (exact arithmetic: *t* = 12.95, *p* < 0.001, Cohen’s d = 2.12; approximate arithmetic: *t* = 7.15, *p* < 0.001, Cohen’s d = 0.64). In the phonological task, *t*-tests were conducted to compare reaction times between the rhyme task and the character matching task, yielding results of *t* = 1.71, *p* = 0.09. Similarly, in the semantic task, *t*-tests were performed to compare reaction times between the semantic task and the pseudo-word-matching task, with results of *t* = −1.49, *p* = 0.14. These findings indicate that participants exhibited similar reaction times under the two conditions in both the phonological and semantic processing tasks. Accuracy and reaction times for each condition are presented in Table 2.

### 3.2. fNIRS Results

#### 3.2.1. Beta Values Results

A 2 × 2 repeated measures ANOVA was conducted on the beta values. The results showed a significant main effect of arithmetic type in the L-SFG (CH16) (*F* (1, 28) = 5.34, *p* = 0.02), with the beta value for exact arithmetic significantly greater than that for approximate arithmetic. A significant main effect of operand size was observed in the middle of the R-SFG (CH4) (*F* (1, 28) = 5.95, *p* = 0.02), with the beta value for large operands significantly greater than that for small operands. Significant interactions were found in CH2, CH3, CH4, CH5, CH10, and CH15 (*Fs* > 4.67, *ps* < 0.05). Further simple effects analysis (FDR corrected) revealed that in the R-SFG (CH2/CH3) and the middle of the R-SFG (CH4), the beta value for large operands was significantly greater than that for small operands under the approximation condition, while no significant difference was observed between these operand sizes under the exact arithmetic condition. The specific results are shown in Table 3 and Table 4 and Figure 5.

#### 3.2.2. Correlation Analysis Results

We attempted to use the rhyme and semantic tasks as localization tasks to identify brain regions associated with phonological and semantic processing. However, these tasks did not yield significant activations in the targeted regions (The marginally significant channel in the phonological task: CH23, *t* = 1.72, *p* = 0.09; For the other phonological tasks, *ts* < 1.45, *ps* > 0.09; Semantic task: *ts* < 1.35, *ps* > 0.05), likely due to the lower sensitivity of fNIRS in detecting brain activity compared to fMRI. However, previous behavioral and neurocognitive studies ([9]; [10]; [47]) have shown that correlations in reaction times or brain activation patterns across tasks can, to some extent, infer associations between different cognitive processing mechanisms. Based on this foundation, the current study analyzed the associations between brain activation patterns during exact and approximate arithmetic tasks and those during semantic and phonological processing.

In exploring the correlation between the beta effect sizes of phonological and semantic tasks and the beta effect sizes of exact arithmetic tasks, we found no significant correlation in the R-SFG (CH2), R-PMC and R-SMA (CH3), R-DLPFC (CH4), right orbitofrontal gyrus (CH5), and L-SFG (CH15) (|*rs*| < 0.21, *ps* > 0.05). Similarly, the beta effect sizes of the phonological task were not significantly correlated with the beta effect sizes of an approximate arithmetic task in these brain regions (|*rs*| < 0.20, *ps* > 0.05). However, it is noteworthy that in the R-PMC and R-SMA (CH3), a significant positive correlation was found between the beta effect sizes of the semantic task and the beta effect sizes of the approximate arithmetic task (*r* = 0.36, *p* = 0.04), as shown in Figure 6.

## 4. Discussion

Utilizing the fNIRS technique, this study investigated cognitive and neural differences between multi-digit exact and approximate arithmetic, as well as their possible association with phonological and semantic processing. Behavioral results revealed significant cognitive differences between the two types of calculations, with exact arithmetic requiring more time, though both showed a significant problem size effect. fNIRS analysis further revealed the neural differences between these calculations, with exact arithmetic eliciting stronger activation in the L-SFG compared to approximate arithmetic, while only approximate arithmetic showed problem size effect in the right hemisphere, including the R-DLPFC, R-SFG, R-PMC and R-SMA. Correlation analysis of brain activities revealed a significant relationship between the problem size effect in approximate arithmetic and semantic processing.

### 4.1. Neural Mechanisms of Exact and Approximate Arithmetic

Earlier studies found that approximate arithmetic activated the parietal and frontal regions, especially the IPS, while exact arithmetic showed greater activation in the left inferior frontal area and AG ([42]). However, our study observed that exact arithmetic significantly activated the L-SFG only and not the AG. This discrepancy may have occurred because the AG is primarily responsible for solving problems through fact retrieval. In our study, the arithmetic conditions for large operands included two-digit numbers, which likely prevented participants from solving problems through simple fact retrieval. As the operand size increased, the role of automatic fact retrieval diminished, and quantity-based processing became more important. In fact, prior findings have revealed that as operand size increased, the activation in the AG decreased ([50]). The AG showed stronger activation during single-digit calculations, which can be solved through fact retrieval than during multi-digit calculations, which cannot be solved directly through retrieval ([22]).

Generally, the left frontal lobe is involved in the demands of working memory and strategy selection, with activation observed in this brain region during relatively complex tasks or multi-digit calculations ([15]; [24]; [37]; [48]). In exact arithmetic, participants must ensure that each operand is processed thoroughly. Thus, language involvement may be indispensable when generating the answer. Furthermore, frontal cortex activation may reflect a greater load on associated working memory and control functions ([26]). Therefore, the fNIRS results may indicate that when arithmetic facts cannot be directly retrieved from memory to solve a problem, individuals need to rely on multi-step calculations. The significant activation of the L-SFG indicates a greater reliance on working memory and sequential processing for multi-step problem-solving. In contrast to approximate arithmetic, which may quickly yield an answer by simplifying the problem, exact arithmetic relies more heavily on working memory. Thus, multi-digit exact arithmetic requires greater memory demands than approximate arithmetic, highlighting the fundamental difference between exact and approximate arithmetic processes.

### 4.2. Problem Size Effect in Exact and Approximate Arithmetic

Numerous studies have observed the problem size effect in exact arithmetic across different operations (addition, subtraction, multiplication, division) and number comparison ([6]; [7]; [12]; [29]; [49]). Consistent with previous research, this study also found the problem size effect in both RTs, showing that larger operands impose a greater cognitive load, resulting in longer reaction times. Additionally, we revealed the problem size effect in approximate arithmetic, which is evident not only at the behavioral level but also at the neural level.

In terms of behavioral results, this study found that the problem size effect in approximate arithmetic is consistent with the findings of [20] ([20]), despite the difference in operation types (i.e., addition and subtraction in this study versus multiplication in Ganor-Stern et al.’s research). This suggests that the problem size effect in approximate arithmetic is stable across different operations. However, some studies found no problem size effect in approximate arithmetic ([33]) or found it to be less pronounced than in exact arithmetic ([42]). This discrepancy may be attributed to differences in the number of digits involved (e.g., single-digit vs. multi-digit). The present study, similar to that of [20] ([20]), employed two levels of operand sizes, thereby identifying the range of sensitivity to operand size in approximate arithmetic. The behavioral results also revealed a significant interaction between arithmetic type and operand size. While the main effect of operand size was significant under both exact and approximate conditions, the reaction time difference between large and small operands was more pronounced in the exact arithmetic condition. This suggests that the cognitive processing associated with operand size differs slightly between arithmetic types, with a greater difference observed in exact arithmetic. This may be related to the calculation strategy required for exact arithmetic, which demands precise results. As operand size increases, more cognitive resources are required, leading to longer reaction times.

In terms of neural findings, the problem size effect in approximate arithmetic was observed primarily in the R-SFG, R-DLPFC, R-PMC, and R-SMA, which aligns in part with previous research. Prior studies have also identified the R-DLPFC, along with the left precentral sulcus and IPS, as sensitive to the problem size effect in approximate arithmetic. Moreover, these regions played a critical role in exact arithmetic as well ([42]). The discrepancy observed in this study might be attributed to the larger range of operands used. Exact arithmetic may be more sensitive to operand size in single-digit calculations, while the behavioral effects of multi-digit exact arithmetic likely involve the coordination of multiple brain regions rather than specific sensitive areas.

### 4.3. Association Between Exact/Approximate Arithmetic and Language Processing

Regarding the association between exact/approximate arithmetic and language processing, our results revealed that brain activities of approximate arithmetic were associated with semantic processing. Previous research involving bilingual individuals or participants with language deficits has demonstrated that language played a special role in exact arithmetic ([14]; [42]; [50]). However, this study found no connection between exact arithmetic and the language processing network, including phonological and semantic processing. One possibility for this discrepancy might be that previous studies often used single-digit numbers as experimental materials, where the relationship between exact arithmetic and language was primarily mediated through fact retrieval ([14]; [42]). In contrast, this study employed multi-digit numbers and aimed to reveal the cognitive and neural mechanisms activated by addition and subtraction calculations. Even with extensive educational training from a young age, most participants cannot memorize all the answers to multi-digit addition and subtraction problems ([21]) and thus find it difficult to solve exact arithmetic by directly retrieving answers from memory. Since this study used multi-digit problems, participants were unable to use fact retrieval strategies and likely relied on multi-step calculations to solve the problems, thus demonstrating a more distant relationship with language processing.

In fact, previous studies investigating the mechanisms of approximate arithmetic have often used the (delayed) verification paradigm ([14]; [30]). In single-digit approximate arithmetic, directly obtaining an accurate answer is difficult to avoid. The difference between the accurate answer and the options primarily involves the mental number line, which entails spatial processing ([14]). In multi-digit approximate arithmetic, the comparison might involve processing the semantic concepts of “large” and “small,” implicating semantic processing and leading to the conclusion that the semantic network is involved in approximate arithmetic ([30]). However, the sensitive region for approximate arithmetic, the R-SMA, is actually associated with sustained attention and response control. Therefore, another possibility for the correlation between approximate arithmetic and semantic processing may be the sharing of cognitive demands involved in both tasks.

Nonetheless, our study offers unique contributions. First, by utilizing the production task, we were able to more objectively reveal the cognitive and neural mechanisms of calculation, minimizing the influence of unrelated factors such as cognitive comparison (in comparison tasks) or inhibitory control (in the delayed verification paradigm). Additionally, by focusing on multi-digit calculations, this study avoided the influence of strategic retrieval commonly seen in single-digit calculations, providing a more accurate reflection of the differences in how exact and approximate arithmetic are processed in the adult brain. These findings suggest that the developmental differences between exact and approximate arithmetic ultimately lead to distinct brain functions in humans. This study highlights the neural mechanisms of exact and approximate arithmetic, offering insights for math education. Exact arithmetic benefits from step-by-step strategies, such as breaking problems into smaller tasks. For approximate arithmetic, activities like range estimation can improve number sense and flexibility.

## 5. Limitations

Although this study provides new insights into the neural mechanisms of exact and approximate arithmetic processing and their potential association with the language network, several limitations should be acknowledged. The relationship between the language tasks and arithmetic tasks was primarily interpreted based on the correlation of regional activities, which represents an exploratory analysis. However, such inferences are limited to the level of correlation and cannot confirm that language tasks directly participate in the processing of mathematical tasks. Therefore, the conclusions drawn here are restricted to uncovering potential associations between the two types of tasks. The findings from the correlation analysis have certain limitations. Although the correlation analysis of brain activation patterns revealed an association between approximate arithmetic and semantic processing, this evidence is indirect and does not clarify whether they share the same neural functional basis. Future research should further explore this relationship through brain functional localization approaches or by manipulating language-related cognitive abilities to observe changes in mathematical problem-solving processes or neural mechanisms, thereby determining the causal link between the two.

Future research could explore incorporating more complex types of arithmetic operations (e.g., fraction calculations, multi-step operations, or multiplication and division) to investigate differences in neural activation across various contexts. Leveraging higher-resolution brain imaging technologies could further uncover the fine-grained neural mechanisms underlying arithmetic cognition, offering valuable insights for educational practices.

## 6. Conclusions

Combining the production paradigm and fNIRS technique, this study examined the cognitive and neural bases of exact and approximate arithmetic, as well as their possible difference and association with language processing. The present results revealed that exact arithmetic not only required more time but also showed a neural dissociation with approximate arithmetic. Additionally, in multi-digit calculations, approximate arithmetic, but not exact arithmetic, showed a closer association with semantic processing.

## Figures and Tables

**Figure 1 behavsci-15-00033-f001:**
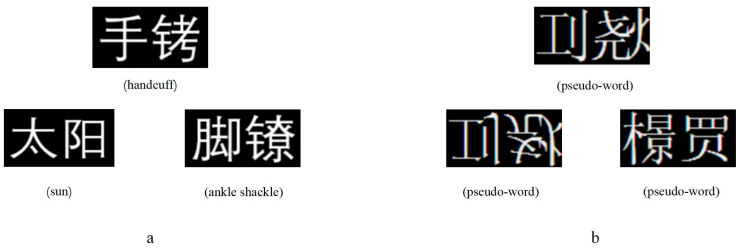
Experimental Materials of the Semantic Processing Task. Note: The stimuli for the semantic task include real and pseudo-words. The first row shows the target stimuli, and the second row shows the two options. In the real-world task, participants choose the word semantically related to the target stimulus (**a**); in the pseudo-word task, they choose the mirror image of the target stimulus (**b**).

**Figure 2 behavsci-15-00033-f002:**
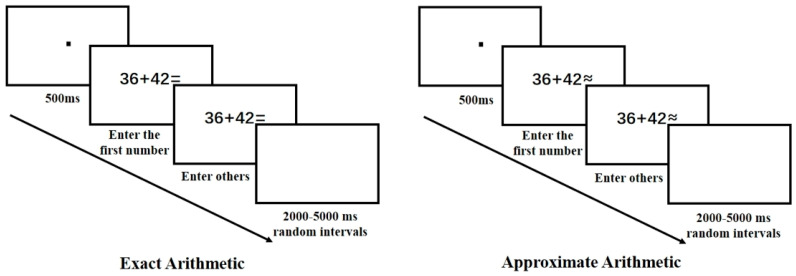
Workflow for Exact and Approximate Arithmetic Tasks. Note: The tasks include a random sequence of addition and subtraction problems with small and large operands.

**Figure 3 behavsci-15-00033-f003:**
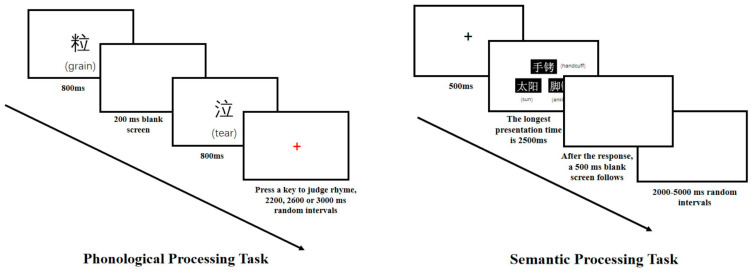
Workflow for Phonological and Semantic Processing Tasks.

**Figure 4 behavsci-15-00033-f004:**
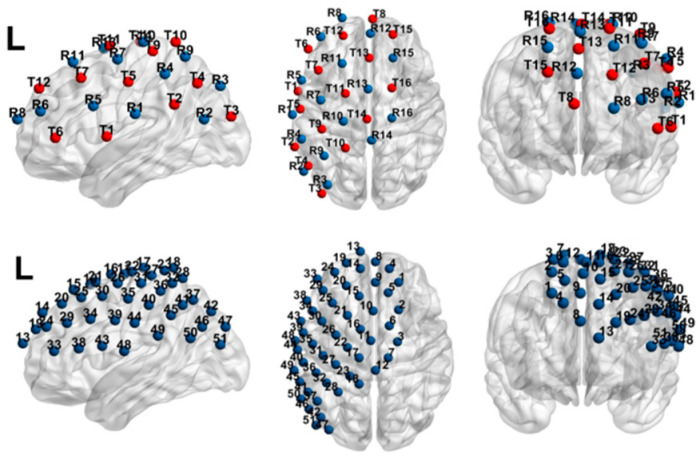
The fNIRS-Based Brain Channel Distribution Map. Note: In the upper figure, red dots represent emitters, and blue dots represent detectors. The lower figure shows the arrangement of the channels formed by the probes.

**Figure 5 behavsci-15-00033-f005:**
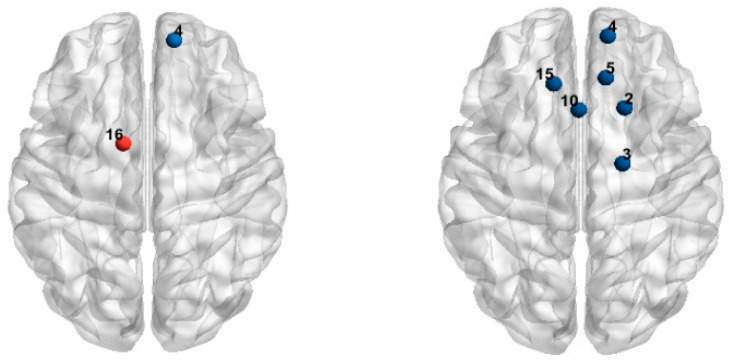
Brain Region Activation Map. Note: In the left figure, red denotes channels with a significant main effect of arithmetic type, and blue denotes channels with a significant main effect of operand size. The right figure shows channels with significant interaction effects.

**Figure 6 behavsci-15-00033-f006:**
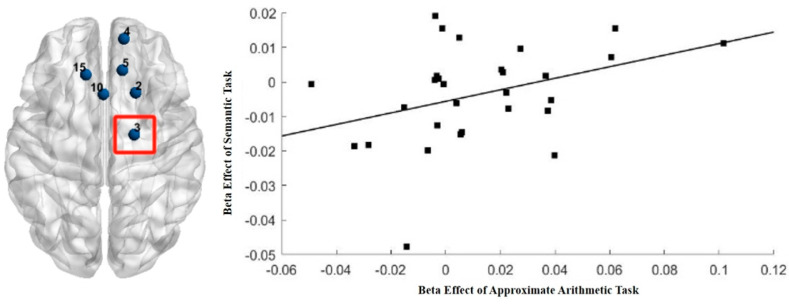
Correlation between semantic task beta effect sizes and approximate arithmetic task beta effect sizes.

**Table 1 behavsci-15-00033-t001:** Examples of Arithmetic Tasks.

Exact Arithmetic		Approximate Arithmetic	
Small Operands	Large Operands	Small Operands	Large Operands
23 + 5 =	36 + 42 =	53 + 4 ≈	31 + 56 ≈
2 + 83 =	52 + 34 =	7 + 32 ≈	61 + 37 ≈
24 − 3 =	97 − 41 =	79 − 8 ≈	89 − 32 ≈
89 − 5 =	73 − 31 =	78 − 6 ≈	94 − 43 ≈

Note: Small operands refer to single-digit ± two-digit operations, and large operands refer to two-digit ± two-digit operations. The same problems were counterbalanced between exact and approximate arithmetic tasks for different participants.

**Table 2 behavsci-15-00033-t002:** Accuracy and Reaction Times (RT) Across Different Tasks and Conditions.

Condition	Accuracy	RT (ms)
Exact Arithmetic with Small Operands	0.96 (0.04)	1730.57 (435.50)
Exact Arithmetic with Large Operands	0.94 (0.04)	2555.22 (663.97)
Approximate Arithmetic with Small Operands	0.89 (0.15)	1552.45 (293.04)
Approximate Arithmetic with Large Operands	0.82 (0.10)	1801.68 (364.72)
Rhyme	0.94 (0.04)	776.36 (365.55)
Character Matching	0.82 (0.05)	631.18 (323.12)
Semantic	0.99 (0.03)	1188.86 (360.38)
Pseudoword Matching	0.98 (0.04)	1319.70 (353.93)

Note: Values in parentheses represent standard deviations (SD).

**Table 3 behavsci-15-00033-t003:** Beta Values for Significant Channels Across Arithmetic Conditions.

Channel (CH)	Condition			
	E1	E2	A1	A2
2	0.018	0.012	0.005	0.016
3	0.016	0.009	0.003	0.015
4	0.012	0.013	−0.001	0.009
5	0.012	0.008	0.005	0.011
10	0.020	0.014	0.011	0.018
15	0.016	0.011	0.007	0.013

Note: E—exact arithmetic, A—approximate arithmetic, 2—large operands, 1—small operands.

**Table 4 behavsci-15-00033-t004:** Results of Interaction and Simple Effects Analysis of the Two Independent Variables.

Channel	MNI Coordinates			Brodmann Area	*F*	η_p_^2^	FDR Corrected	
(CH)	(x,y,z)			(BA)			E2-1	A2-1
2	21	19	67	BA8-R	5.92 *	0.17	0.16	0.03
3	20	−9	76	BA6-R	4.67 *	0.14	0.22	0.04
4	13	55	45	BA9-R	6.98 *	0.20	0.80	0.003
5	12	34	61	BA8-L	5.48 *	0.16	0.16	0.08
10	−1	18	66	BA6-L	6.22 *	0.18	0.06	0.11
15	13	31	62	BA8-L	5.75 *	0.17	0.18	0.07

Note: E—exact arithmetic, A—approximate arithmetic, 2—large operands, 1—small operands. * *p* < 0.05.

## Data Availability

The data presented in this study are available on request from the corresponding author. The data are not publicly available due to privacy.

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
