# Peer review of "Cognitive and Neural Differences in Exact and Approximate Arithmetic Using the Production Paradigm: An fNIRS Study"

_behavsci, 2025, doi:10.3390/bs15010033_

Round 1
Reviewer 1 Report
Comments and Suggestions for Authors
The study examined the cognitive and neural differences between exact and approximate arithmetic processing, utilizing fNIRS technology and behavioral analysis. Besides, it also explored the role of phonological and semantic function during arithmetic processing, to further understanding the cognitive mechanisms. Generally speaking, this article is well-written, effectively providing an in-depth study of exact and approximate arithmetic processing. The discussion section summarizes the experimental results and compares them with existing literature, offering reasonable explanations. However, I recommend the following improvements to enhance the quality and persuasiveness of the manuscript:
Point 1: In the task setting, exact and approximate arithmetic were only distinguished by symbols. Would there be confusion during the test? Could the participants understand the task rules?
Point 2:This paper introduces the exact and approximate arithmetic experimental materials in detail. Can the developed exact and approximate arithmetic experimental materials distinguish between these two processes? Please intrudce this issue further.
Point 3:While the article outlines the research gap concerning the ecological validity of previous paradigms and the need to study multi-digit arithmetic, the discussion on how the production paradigm overcomes these limitations could be further emphasized. It would be beneficial for the authors to more explicitly highlight how their approach is novel and how the production paradigm addresses the shortcomings of the verification and comparison tasks. This would enhance the clarity of the study's contribution to the field.
Point 4:The manuscript briefly mentions the potential educational implications of the findings for teaching arithmetic in primary and secondary education. I suggest expanding this section to include a more detailed discussion about how the knowledge of the neural mechanisms of exact and approximate arithmetic could inform teaching practices. Providing concrete examples of how these findings could be applied in educational settings would increase the practical relevance of the study.
Point 5:Regarding the neural mechanisms of the size effect, the authors offer detailed discussions at both the behavioral and neural levels, especially the activation of the right-sided brain regions (e.g., right dorsolateral prefrontal cortex, right superior frontal gyrus) in approximate arithmetic. However, the paper lacks a detailed explanation of the neural mechanisms behind the size effect in exact arithmetic. I recommend that the authors provide more discussion on the neural mechanisms involved in the size effect during exact arithmetic, exploring whether multi-digit calculations engage additional brain regions and how they relate to working memory and cognitive control.
Point 6: Association between Exact Arithmetic and Language Processing
The article indicates that no significant association was found between multi-digit exact arithmetic and the language processing network, which differs from previous studies mainly using single-digit problems. The authors attribute this difference to the inability to solve multi-digit problems through fact retrieval strategies. This explanation is reasonable, but I suggest further exploration of whether other potential factors, such as educational background, types of arithmetic, or task design, may play a role. Additionally, the authors could discuss whether it is possible to observe a connection between exact arithmetic and language processing under different conditions (e.g., different types of language tasks or more complex arithmetic tasks).
Point 6: This study used G Power to estimate sample size, however, for two way within-subject design, you can using PANGE (https://jakewestfall.shinyapps.io/pangea) to calculate the sample size.
Point 7: The system included 16 detectors and 16 receivers, you mean 16 sources and 16 detectors?
Point 8: For the Figure legends, please present more detalied information ahout the figure and the main results. Such as, “Figure 5. Correlation Analysis Diagram”, they are too brief.
Comments on the Quality of English LanguageThe structure and logic of the study is reasonable and the writting is clear.
Author Response
Please see the attachment. Thank you for all your advice.

Reviewer 2 Report
Comments and Suggestions for Authors
Thanks for the opportunity to review this paper. This fNIRS study investigates neural correlates of approximate and exact arithmetic with typical, Chinese speaking, adults, while also considering the role of language-related systems. Unlike previous studies on the topic, the task design is posed as involving a more naturalistic and authentic setting. As a numerical cognition researcher, I find the topic and the research questions as of interest to the field. Use of fNIRS is argued to enable a task design that would be difficult to use in fMRI or EEG, due to movement artifacts. The study has the potential to make a good contribution to the literature but needs some improvements. I list the major issues below, with more minor issues listed underneath.
The arithmetic task involves addition and subtraction questions with small and large operands, for both the approximate and exact task conditions. Even though two different operations are used (add vs. sub), this factor is not included in the analysis. I would appreciate a justification for this choice. Unlike most other neuroimaging studies, where a validation task is used, the task used in this study requires participants to type their answers using the keyboard and this is argued to make the task more naturalistic. While this task design removes the additional processing required for choosing between two alternative answers, it requires a more complicated motor response generation, due to entry of the answer using the keyboard. I believe, this is an acceptable compromise, since fNIRS is not affected by motor artifacts like other modalities. Though, to what extent this is more authentic/naturalistic than validation is up for debate, since in most real-life problem-solving scenarios, students are expected to solve multiple choice questions, which is a form of validation. I would appreciate a brief discussion on why the authors think that entering the response with the keyboard is a more naturalistic choice. I think this would help the readers better understand the task design.
I recommend the authors to consider reverse inference issues in both the introduction and their interpretation of the results in the discussion. To state the obvious, reference inference is deducing mental states based on observed activations. For example, observing partial overlap between a set of areas associated with language processing and areas observed in a mathematical task alone, does not warrant any conclusions about the participation of language processing in the math task. This fallacy is more obvious if we, hypothetically, do the reverse; localize a set of regions with a math task and deduce that mathematical processing is involved in a language task, because of neural overlap.
I found the justifications for the statistical analysis a bit cryptic. Please explain the rationale for all procedures, for example finding differences in beta values, and running correlation analyses, in addition to the ANOVAs.
Minor Issues:
Abstract. “while approximate arithmetic exhibited effects in the right hemisphere” Which effects?
34-36. Given the body of work on neural correlates of approximate vs exact arithmetic, I don’t think this statement is necessarily true. Also, the papers cited (3-7) are not really papers focusing on approximate vs exact arithmetic. I included some example papers on the topic, which are ignored in the manuscript. I suggest the authors conducting a more comprehensive literature review.
Kucian, K., von Aster, M., Loenneker, T., Dietrich, T., & Martin, E. (2008). Development of neural networks for exact and approximate calculation: A FMRI study. Developmental neuropsychology, 33(4), 447-473.
Davis, N., Cannistraci, C. J., Rogers, B. P., Gatenby, J. C., Fuchs, L. S., Anderson, A. W., & Gore, J. C. (2009). The neural correlates of calculation ability in children: an fMRI study. Magnetic resonance imaging, 27(9), 1187-1197.
96-110. Problem of reverse inference. Seeing IPS activation for a task does not necessarily mean that magnitude processing is taking place. The same goes for regions considered in the “language network” All of these regions participate in a myriad of functions. We cannot infer function, by activation alone.
186-187. Why not just call it a 2x2x2 design, given that arithmetic task (add vs. sub) is also a factor with two levels?
194-206. What does “neural association” mean? I am assuming it refers to neural overlap. Does neural overlap also mean functional overlap? In other words, if there is more neural overlap between some language task and a math task, compared to another math task; can we really infer that the former math task involves more language processing? Please reconsider this approach, based on issues with reverse inference.
209-2011. I don’t think I understand how the power analysis was conducted for the neuroimaging analysis. Was this conducted for the behavioral data?
Please report participants’ native language, since this study focuses on arithmetic and language processing.
219-220. Why is arithmetic operation (add vs sub) not another factor, making it a 2x2x2 design, since half of the questions are addition and the other half are subtraction? Is it because the differences between operations is not of interest?
269—286. I could not understand the task design for the arithmetic task. What do “enter the first number” and “enter others” mean? Please explain the task design in a clear manner.
308-309. Please explain the rationale for the distribution of the sensors. It seems like they are mostly left-lateralized (excluding inferior frontal and temporal), with some right-lateralized sensors along the anterior cingulate.
316. What is the criteria for “unreasonable trials” for all tasks? Also, please indicate how many trials for excluded on average (with SD values).
326. Please include the citation for NIRS_SPM. This is how open source and free research software get the credit they deserve.
336-341. As far as I understand, the rhyming and the semantic tasks are used as localizers for the language network. Please clarify.
351–360. Please clarify the procedure for subtracting beta values. Why is this done?
When talking about the 2x2 ANOVAs, please include what the factors are.
399-410. Please explain why correlation analyses are conducted. In general, it helps to first provide an explanation for why a specific analysis is conducted.
In regard to the analysis conducted, I am a bit confused why a 2x2 ANOVA on the arithmetic fNIRS result, followed by correlation analysis is conducted. Please correct me if I am wrong, but I expected the language tasks to be used as localizers for the language network and then the arithmetic analysis to be constrained by the regions (sensors) pointed by the language localizer. Please justify the analysis conducted.
Comments on the Quality of English Language
Please proof-read the manuscript and improve language use for flow and clarity.
Author Response

(The authors gave the same response as above.)

Reviewer 3 Report
Comments and Suggestions for Authors
Review of behavsci-3220899
Title: Cognitive and Neural Differences in Exact and Approximate Arithmetic using the Production Paradigm: An fNIRS Study
Summary:
This study investigated the cognitive and neural mechanisms of exact and approximate
arithmetic using fNIRS technology during natural calculation processes. With the production paradigm, this study employed a 2 (exact arithmetic vs. approximate arithmetic) × 2 (large operands vs. small operands) within-subject experimental design using integer addition and subtraction to explore the cognitive-neural mechanisms of multi-digit exact and approximate arithmetic and their differences in an fNIRS experiment. The fNIRS results revealed differences in the neural bases underlying these two arithmetic processes. Additionally, larger operands registered more brain activities in the R-DLPFC (right dorsolateral prefrontal cortex), R-SFG (right superior frontal gyrus), and PMC and SMA (pre- and supplementary motor cortexes) compared to smaller operands in approximate arithmetic. Moreover, correlation analysis found a significant correlation between approximate arithmetic and semantic processing in the R-PMC and R-SMA (right pre- and supplementary motor cortexes). These findings suggest a neural dissociation between exact and approximate arithmetic, with exact arithmetic processing showing a dominate role in the left hemisphere, while approximate arithmetic processing was more sensitive in the right hemisphere.
Overall assessment:
This is an interesting study in exact and approximate arithmetic However, as the comments below indicate, I have some concerns regarding introduction, methods, results, and discussion, which should be addressed before recommending publication.
1) Materials and Methods: page 5, lines 226-2227.” After excluding participants with excessive artifacts detected during preprocessing, 29 valid data sets remained”. Is it possible to be specific in order to indicate how much the artifacts exceed?
2) Materials and Methods: page 7, lines 291. Please report specific experimental material parameters, including the size of the fixation, the size of the stimulus, and the color of the background.
3) Materials and Methods: page 8, lines 322. Is it possible to add a figure to show the phonological processing task and the semantic processing task?
4) Materials and Methods: page 9, lines 394. Please specify how the correlation is calculated, is it a Pearson correlation?
5)Materials and Methods: page 8, lines 348. Please specify the behavioral analysis methods for the phonological and semantic tasks.
6) Results: page 9, lines 401. I didn't see specific response time and accuracy data about each condition, please add it. Additionally, I did not see the analysis regarding the phonological and semantic tasks. If possible, could you put a figure to show the behavioral data?
7) Results: page 10, lines 413. As with the behavioral results, please specify the Beta value for each condition. Also, regarding the Beta value results for the phonological and semantic tasks I have not seen them, please add them.
8) Results: page 10, lines 434. Please add a specific analysis of the correlation results, the significance of each region, and the correlation coefficients.
9) Discussion: page 11, lines 517. “In contrast, this study employed multi-digit numbers and aimed to reveal the cognitive and neural 518 mechanisms activated by multiplication calculations.” Was it a clerical error? The experimental conditions were not manipulated for multiplication. Also, I see that the calculations in the Table 1 example do not utilize multiplication in any way, are there any examples added to illustrate the discussion here?
10) Figures and labels throughout the text need to be standardized. For example, in Figure 1, please label a and b for real and pseudo-words and explain them in the notes. The same applies to other Figures.
11) Just a small suggestion. I understand the logic of the quote but is the quote too long? Please make the quote more concise if you can.
Comments on the Quality of English LanguageThe language is clear, with a few places where sentences are too long and need to be optimized.
Reviewer 4 Report
Comments and Suggestions for Authors
The paper presents a valuable contribution to the understanding of cognitive and neural mechanisms in arithmetic processing. The combination of behavioral and neuroimaging data provides a comprehensive view of the differences between exact and approximate arithmetic. With minor revisions for clarity and detail, it has strong potential for publication.
Suggestions for Improvement
- It is suggested to include demographic details of participants: Lack of detailed demographic information (e.g., age range, gender distribution) limits the ability to assess the representativeness of the sample.
- The criteria for excluding participants or trials due to artifacts need clarification to ensure transparency in data handling.
- While the use of repeated measures ANOVA is appropriate, the paper should provide more detail on effect sizes and power analyses to substantiate the statistical findings. This would help readers understand the practical significance of the results.
- The paper mentions a size effect in reaction times but does not adequately explore how this effect differs between exact and approximate arithmetic in detail. More analysis could strengthen the findings.
- Some claims about the cognitive processes underlying exact and approximate arithmetic may be too broad. The discussion could benefit from a more nuanced interpretation of how specific brain regions relate to different types of arithmetic tasks.
- The paper lacks a dedicated limitations section. Addressing potential confounding variables, such as participants' prior experience with arithmetic tasks or differences in cognitive strategies, would enhance the study's credibility.
The writing is very good, no any difficulty to read the manuscript.
Round 2
Reviewer 2 Report
Comments and Suggestions for Authors
I appreciate the authors taking the time to address the comments in the previous round of reviews. The authors addressed some of the issues that were raised. However, I am troubled with the path they followed with the localization analysis. As far as I understand, localization analysis with the language tasks were conducted, though the results showed no significance and were therefore omitted, yet beta values from these sites were still used for the correlation analysis. I reflected on why this is an issue below.
Comments 9: 209-2011. I don’t think I understand how the power analysis was conducted for the neuroimaging analysis. Was this conducted for the behavioral data?
I don’t think this question was answered. For which comparison and sensor site was the effect size estimated in the power analysis? Based on which previous study, or what rationale, was the effect size “f = 0.25” chosen? Unlike behavioral studies, power analysis in neuroimaging studies is more complex since variance and mean activation vary depending on the brain region (for fMRI) or sensor site (for fNIRS). In addition, again unlike behavioral studies, neuroimaging studies involve multiple ROIs, increasing the number of multiple comparisons. The fact that rhyming and semantic processing tasks could not localize the language network well exemplifies how the power calculated for behavioral effects here, may not be enough for neuroimaging effects.
There is really no consensus on how we should conduct power analysis in neuroimaging, some studies suggesting use of pilot data, others conducting posteriori power analysis for specific ROIs and statistical tests. Overall, I think it is important to detail whatever path is followed. I don’t expect the authors to resolve the issues involved with power analysis with fNIRS data, but the method laid out here works only for behavioral data and not suitable for fNIRS.
Comment 13. Please include this explanation in the manuscript.
Comments 16: If these regions are not showing significant results for rhyming and semantic tasks then why is it reasonable to extract beta values for the correlation analysis? Also, I do not think it is reasonable to not report the localization analysis results; they should be reported in the paper. Trying different types of analysis, reporting only the significant results, and changing the analysis pipeline to sway away from null results increases TYPE 1 error rates. These issues are extensively discussed on studies on the replication crisis in neuroimaging and psychology.
Comments 19. “if certain brain regions exhibit significant semantic or phonological processing features during language tasks and simultaneously show activation patterns closely related to the operand size effect during arithmetic tasks, it may suggest that these regions share neural mechanisms or cognitive resources across the two types of tasks.” But, in your response to Comments 19, you indicated that these regions don’t show significance for the semantic and phonological processing tasks. Isn’t this an issue?
